# Combination Regimens of Favipiravir Plus Interferon Alpha Inhibit Chikungunya Virus Replication in Clinically Relevant Human Cell Lines

**DOI:** 10.3390/microorganisms9020307

**Published:** 2021-02-02

**Authors:** Evelyn J. Franco, Xun Tao, Kaley C. Hanrahan, Jieqiang Zhou, Jürgen B. Bulitta, Ashley N. Brown

**Affiliations:** 1Institute for Therapeutic Innovation, Department of Medicine, College of Medicine, University of Florida, Orlando, FL 32827, USA; e.franco@ufl.edu (E.J.F.); Kaley.Hanrahan@medicine.ufl.edu (K.C.H.); 2Department of Pharmaceutics, College of Pharmacy, University of Florida, Orlando, FL 32827, USA; tealingsxun@ufl.edu; 3Department of Pharmacotherapy & Translational Research, College of Pharmacy, University of Florida, Orlando, FL 32827, USA; zhou.jieqiang@cop.ufl.edu (J.Z.); JBulitta@cop.ufl.edu (J.B.B.)

**Keywords:** chikungunya virus, favipiravir, interferon-alpha, combination therapy, mathematical modeling

## Abstract

Chikungunya virus (CHIKV) is an alphavirus associated with a broad tissue tropism for which no antivirals or vaccines are approved. This study evaluated the antiviral potential of favipiravir (FAV), interferon-alpha (IFN), and ribavirin (RBV) against CHIKV as mono- and combination-therapy in cell lines that are clinically relevant to human infection. Cells derived from human connective tissue (HT-1080), neurons (SK-N-MC), and skin (HFF-1) were infected with CHIKV and treated with different concentrations of FAV, IFN, or RBV. Viral supernatant was sampled daily and the burden was quantified by plaque assay on Vero cells. FAV and IFN were the most effective against CHIKV on various cell lines, suppressing the viral burden at clinically achievable concentrations; although the degree of antiviral activity was heavily influenced by cell type. RBV was not effective and demonstrated substantial toxicity, indicating that it is not a feasible candidate for CHIKV. The combination of FAV and IFN was then assessed on all cell lines. Combination therapy enhanced antiviral activity in HT-1080 and SK-N-MC cells, but not in HFF-1 cells. We developed a pharmacokinetic/pharmacodynamic model that described the viral burden and inhibitory antiviral effect. Simulations from this model predicted clinically relevant concentrations of FAV plus IFN completely suppressed CHIKV replication in HT-1080 cells, and considerably slowed down the rate of viral replication in SK-N-MC cells. The model predicted substantial inhibition of viral replication by clinical IFN regimens in HFF-1 cells. Our results highlight the antiviral potential of FAV and IFN combination regimens against CHIKV in clinically relevant cell types.

## 1. Introduction

Chikungunya virus (CHIKV), a mosquito borne alphavirus endemic to Africa and Asia, first emerged in the Western Hemisphere in 2013 [1]. Since then, it swiftly spread throughout the Caribbean and Americas where it has caused an excess of 2 million infections [2,3,4,5]. CHIKV is associated with high morbidity rates as only 15% of patients are asymptomatic [6]. Acute CHIKV infection is characterized by debilitating joint and muscle pain, fever, headache, and maculopapular rash; a considerable percentage of patients experience chronic symptoms including recurrent and persistent arthralgia months to years following acute infection [7,8]. Neurologic manifestations including seizure, encephalitis, encephalopathy, and Guillain-Barré syndrome have been reported and tend to be more likely in severe cases of CHIKV infection [6,8,9]. Despite the potential for future outbreaks and high morbidity rate associated with infection, there are currently no antivirals or vaccines approved to treat or prevent CHIKV infection.

The wide range of clinical symptoms in human CHIKV infections is due to the broad tissue and cellular tropism (a virus’s ability to infect a particular cell or tissue type [10]) exhibited by the virus which targets multiple tissues and cell types during infection [11,12]. Chikungunya virus also has a broad cellular tropism in vitro, as a wide variety of human and non-human cell lines are permissive to infection and allow for robust viral replication [13,14]. In our previous work, we have shown that the susceptibility of CHIKV to antiviral therapy is highly variable between cell lines [13]. This underscores the crucial need to select the most appropriate cell types for non-clinical antiviral evaluations to facilitate translation to man. Due to these considerations, we selected three clinically relevant cell lines derived from human tissues to conduct anti-CHIKV evaluations using agents that exhibit broad-spectrum antiviral activity. HT-1080 (human fibrosarcoma) cells are a connective tissue cell line that were selected due to the acute and potentially recurrent arthralgia associated with CHIKV infection. SK-N-MC (human neuroepithelioma) cells were utilized because neuronal cells may be targeted by CHIKV in severe cases of infection. Finally, HFF-1 (human fibroblast) cells are skin cells that represent the first tissue type encountered by virus particles upon mosquito transmission.

Favipiravir (FAV), ribavirin (RBV), and interferon-alpha (IFN) are three licensed agents that exhibit broad spectrum activity against multiple RNA viruses. FAV is an orally available nucleoside RNA-dependent RNA polymerase inhibitor that is approved for the treatment of human influenza in Japan [15] and was evaluated as a potential therapeutic strategy against Ebola virus during the 2014 West African outbreak [16]. RBV is also an orally available antiviral agent that exerts antiviral activity through a variety of proposed mechanisms of action [17] including, acting as a nucleoside RNA-dependent RNA polymerase inhibitor, inhibiting inosine monophosphate dehydrogenase (IMPDH), and immunomodulation [18]. Both FAV and RBV require phosphorylation by host cell kinases into their respective active moieties before they can exert antiviral activity. FAV must be triphosphorylated into its active moiety, FAV-RTP [19,20], whereas RBV can be effective as RBV monophosphate to inhibit IMPDH or as RBV triphosphate which interacts with the virus RNA-dependent RNA polymerase to interfere with viral RNA replication [18]. Finally, IFN is an immunomodulating agent that is approved for use as either monotherapy or in combination with RBV for the treatment of chronic hepatitis C virus (HCV) infection. IFN in combination with RBV was traditionally considered the standard of care for HCV until the advent and approval of direct acting antiviral agents. IFN acts via binding to IFN cell surface receptors leading to the induction of an antiviral state in the cell, making that cell refractory to viral infection [21,22].

Here, we aimed to evaluate FAV, RBV, and IFN as monotherapy and in combination using cell lines derived from human tissue that are clinically relevant to human CHIKV infection [12]. We then further analyzed the clinical potential of the most effective regimens via mathematical modeling using data obtained from the in vitro antiviral evaluations. This translational approach is well suited to identify the most promising antiviral agents and their combinations as a treatment strategy to combat CHIKV. Moreover, this translational approach allows us to predict optimal dosage regimens that maximize viral suppression at non-toxic concentrations in several clinically relevant cell types.

## 2. Materials and Methods

### 2.1. Cell Lines

HT-1080 (ATCC CCL-121) and SK-N-MC (ATCC HTB-10) cells were maintained in MEM (Corning Cellgro; Manassas, VA, USA) supplemented with 10% fetal bovine serum (FBS) (Sigma Aldrich; St. Louis, MO, USA), 1% Penicillin-Streptomycin solution (HyClone; Logan, UT, USA), 1% sodium pyruvate (Hyclone; Logan, UT, USA), and 1% non-essential amino acids solution (Hyclone; Logan, UT, USA). HFF-1 (ATCC SCRC-1041) cells were maintained in Dulbecco’s modified Eagle medium (DMEM), supplemented with 15% FBS (Sigma Aldrich; St. Louis, MO, USA) and 1% penicillin-streptomycin solution (HyClone; Logan, UT, USA). Cells were incubated at 37 °C, 5% CO_2,_ and split twice weekly.

### 2.2. Virus

The vaccine strain of CHIKV (181/clone 25) was obtained from Biodefense and Emerging Infections Research Resources Repository (BEI Resources, Manassas, VA, USA). Viral stocks were prepared as previously described [14].

### 2.3. Antivirals

Favipiravir was obtained from MedKoo Biosciences Inc. (Morrisville, NC, USA), and human interferon-alpha subtype 2a from PBL assay science (Piscataway, NJ, USA). Drug stocks were prepared as previously described [23].

### 2.4. Drug Assays

For monotherapy drug assays, six well plates were seeded with HT-1080, SK-N-MC or HFF-1 cells. To maintain comparable viral replication kinetics between cell lines and account for variability in host cell permissiveness to infection, cells were infected at varying multiplicities of infection (0.1 for HT-1080; 0.01 for SK-N-MC; and 0.001 for HFF-1 cells). Virus was allowed to adsorb onto cells for one hour then the viral inoculum was removed and monolayers were washed twice with PBS to remove unbound virus. Following infection, 3 mL of drug-containing medium (FAV, RBV, or IFN) at concentrations ranging from 0 to 157.10 µg/mL FAV, 0 to 1000 µg/mL RBV, or 0 to 10,000 IU/mL IFN were added to wells. Plates were maintained at 37 °C and 5% CO_2_ for three days. Viral supernatant was sampled daily and clarified by high-speed centrifugation, samples were frozen at −80 °C. Infectious virus was quantified by plaque assay on Vero cells as described previously [14].

For combination drug assays, HT-1080, SK-N-MC, or HFF-1 cells were seeded into 6-well plates and infected as described above for the monotherapy evaluations. All concentrations of FAV ranging from 0 to 157.1 µg/mL and IFN 0 to 10,000 IU/mL were assessed either alone or in combination. Plates were maintained at 37 °C and 5% CO_2_ for two days. Viral supernatant was sampled on day 2 post-infection and processed as described above. Viral burden was quantified by plaque assay on Vero cells [14].

### 2.5. Mathematical Modeling and Simulation

*Mechanism-Based Pharmacodynamic Model:* Previously, we published a mechanism-based pharmacodynamic model (MBM) to describe the inhibitory effects exerted by FAV, IFN and RBV as monotherapy and of the three 2-drug combinations against Zika virus replication [23]. In this study, we refined this MBM to characterize the antiviral effect of these drugs in mono- and combination therapies against CHIKV infection. Our MBM was composed of compartments representing uninfected and infected host cells, as well as extracellular and intracellular virus.

Extracellular chikungunya virus (V_extra_) infected uninfected host cells (U) to form the initial stage of infected host cells (I_i1_). The infection followed a second-order process with the infection rate constant k_infect_. We then utilized a series of five transit compartments (I_i1,_ I_i2,_ I_i3,_ I_i4,_ and I_i5_; linked by a first-order transit rate constant k_tr_) to mimic infected host cells that died at the time of virus egress (i.e., after a delay caused by 5 k_tr_ steps). Uninfected and infected cells were modeled to not replicate in this static plaque assay system. The IFN exerted its inhibitory effects extracellularly. While IFN was modeled to not block the virus binding to the surface receptors on host cells, IFN protected the host cells from being transformed to infected host cells. When host cells were treated with RBV, the cells died via a first-order death-rate constant (k_cytotox_) which was stimulated by a cytotoxic effect of RBV. The cytotoxic effect of RBV affected host cells (infected and uninfected). Due to the killing of infected host cells, loss due to RBV-related toxicity was also applied to the differential equations for intracellular virus. This intracellular virus was immature and, therefore, could not form infectious extracellular virus. The RBV mediated cytotoxicity was modelled via a stimulatory Hill-equation with maximum extent of stimulation (kmax_RBV_) and the RBV concentration (SC_50,RBV_) causing 50% of Smax_RBV_ (Hill coefficient fixed to 1). The differential equations for U and I_i1_ to I_i5_ were:(1)$$\frac{d\;U}{\mathrm{dt}}=-k_{\mathrm{infect}}\cdot \left(1-{\mathrm{INH}}_{\mathrm{IFN}}\right)\cdot V_{\mathrm{extra}}\cdot {U\;-\;k}_{\mathrm{cytotox}}\cdot U\;\;\;\;\mathrm{IC}:\;10^{6.3}\;\mathrm{cells}$$
(2)$$\frac{d\;I_{i1}}{\mathrm{dt}}=k_{\mathrm{infect}}\cdot \left(1-{\mathrm{INH}}_{\mathrm{IFN}}\right)\cdot V_{\mathrm{extra}}\cdot {U\;-\;k}_{\mathrm{cytotox}}\cdot I_{i1}-k_{\mathrm{tr}}\cdot I_{i1}\;\;\;\;\mathrm{IC}:\;{\mathrm{IC}}_{I1}$$
(3)$$\frac{d\;I_{i2}}{\mathrm{dt}}=k_{\mathrm{tr}}\cdot I_{i1}-k_{\mathrm{cytotox}}\cdot I_{i2}\;-\;k_{\mathrm{tr}}\cdot I_{i2}\;\;\;\;\;\;\;\;\mathrm{IC}:\;0$$
(4)$$\frac{d\;I_{i3}}{\mathrm{dt}}=k_{\mathrm{tr}}\cdot I_{i2}-k_{\mathrm{cytotox}}\cdot I_{i3}\;-\;k_{\mathrm{tr}}\cdot I_{i3}\;\;\;\;\;\;\;\;\mathrm{IC}:\;0$$
(5)$$\frac{d\;I_{i4}}{\mathrm{dt}}=k_{\mathrm{tr}}\cdot I_{i3}-k_{\mathrm{cytotox}}\cdot I_{i4}\;-\;k_{\mathrm{tr}}\cdot I_{i4}\;\;\;\;\;\;\;\;\mathrm{IC}:\;0$$
(6)$$\frac{d\;I_{i5}}{\mathrm{dt}}=k_{\mathrm{tr}}\cdot I_{i4}-k_{\mathrm{cytotox}}\cdot I_{i5}\;-\;k_{\mathrm{tr}}\cdot I_{i5}\;\;\;\;\;\;\;\;\mathrm{IC}:\;0$$

The initial condition (IC) for uninfected host cells was set to the targeted inoculum of 10^6.3^ cells/mL. The IC was estimated for infected host cells. The cytotoxicity of RBV was described by a Hill-model for k_cytotox_.
(7)$$k_{\mathrm{cytotox}}=\frac{{\mathrm{kmax}}_{\mathrm{RBV}}\cdot C_{\mathrm{RBV}}}{C_{\mathrm{RBV}}+{\mathrm{SC}}_{50}{}_{,\mathrm{RBV}}}$$

*Viral replication:* Infected host cells generated new intracellular virus (V_i1_) with a synthesis rate constant (ksyn). We synchronized the timing of intracellular virus and infected host cells in order to have infected host cells die (i.e., I_5_ being lost) at the time of virus egress from the last intracellular virus compartment (V_i5_). This was achieved by implementing a rapid equilibrium with an equilibration rate constant (k_eq_: 100 h^−1^; equivalent to a 0.42 min half-life) and by multiplying the k_syn_ by 100. A series of five transit compartments (V_i1,_ V_i2,_ V_i3,_ V_i4,_ and V_i5_; linked by the same first-order transit rate constant [k_tr_] as described above) was used to characterize the different stages of intracellular virus maturation [24]. As described above, RBV-related toxicity to host cells indirectly resulted in a loss of immature intracellular virus. The differential equations for the five intracellular virus compartments (V_i1_ to V_i5_; all initial conditions zero) were:(8)$$\frac{d\;V_{i1}}{\mathrm{dt}}=\mathrm{ksyn}\cdot 100\cdot I_{i1}-k_{\mathrm{eq}}\cdot V_{i1}-k_{\mathrm{tr}}\cdot V_{i1}\;-{\;k}_{\mathrm{cytotox}}\cdot V_{i1}$$
(9)$$\frac{d\;V_{i2}}{\mathrm{dt}}=k_{\mathrm{tr}}\cdot \left(V_{i1}-V_{i2}\right)\;-{\;k}_{\mathrm{cytotox}}\cdot V_{i2}$$
(10)$$\frac{d\;V_{i3}}{\mathrm{dt}}=k_{\mathrm{tr}}\cdot \left(V_{i2}-V_{i3}\right)\;-{\;k}_{\mathrm{cytotox}}\cdot V_{i3}$$
(11)$$\frac{d\;V_{i4}}{\mathrm{dt}}=k_{\mathrm{tr}}\cdot \left(V_{i3}-V_{i4}\right)\;-{\;k}_{\mathrm{cytotox}}\cdot V_{i4}$$
(12)$$\frac{d\;V_{i5}}{\mathrm{dt}}=k_{\mathrm{tr}}\cdot \left[\left(1-\mathrm{INH}\right)\cdot V_{i4}-V_{i5}\right]\;-{\;k}_{\mathrm{cytotox}}\cdot V_{i5}$$

The inhibitory drug effect term (INH) for RBV and FAV is described below. Mature intracellular virus then exits compartment (V_i5_) to the extracellular space (V_extra_) where it is subject either to a first-order loss (rate constant: k_loss, Vextra_), or consumed to form the infected host cells (k_infect_). The inhibitory effect of IFN was not present in the differential equation for Vextra, since IFN was modelled not to inhibit binding of extracellular virus to uninfected host cells. At the initiation of our in vitro experiments, approximately 10^3.3^ plaque-forming units/mL (PFU/mL) CHIKV was loaded into the system. Most extracellular virus was assumed to rapidly infect host cells or to be washed away after initial incubation. However, some virus remained in the extracellular space. Therefore, we estimated the initial condition for V_extra_ to represent the residual extracellular virus at assay initiation. The differential equation was:(13)$$\frac{d\;V_{\mathrm{extra}}}{\mathrm{dt}}=k_{\mathrm{tr}}\cdot V_{i5}-k_{\mathrm{loss},\mathrm{Vextra}}\cdot V_{\mathrm{extra}}-k_{\mathrm{infect}}\cdot V_{\mathrm{extra}}\cdot U\;\;\;\mathrm{IC}:\;{\mathrm{IC}}_{\mathrm{Vextra}}$$

*Drug effect:* The IFN exhibits an antiviral effect by preventing CHIKV to transform uninfected host cells to infected host cells. Our MBM described the inhibitory effect of IFN through an inhibitory Hill function with a maximum extent of inhibition (Imax_IFN_; fixed to 1) and the IFN concentration (C_IFN_) causing 50% of Imax (IC_50,IFN_).
(14)$${\mathrm{INH}}_{\mathrm{IFN}}=\frac{{\mathrm{Imax}}_{\mathrm{IFN}}\cdot C_{\mathrm{IFN}}{}^{{\mathrm{Hill}}_{\mathrm{IFN}}}}{C_{\mathrm{IFN}}{}^{{\mathrm{Hill}}_{\mathrm{IFN}}}+{\mathrm{IC}}_{50}{{}_{,\mathrm{IFN}}}^{{\mathrm{Hill}}_{\mathrm{IFN}}}}$$

FAV and RBV inhibit viral replication and maturation in infected host cells. Their effect was modeled as inhibition of transit from compartment V_i4_ to V_i5_ by an inhibitory Hill function. The FAV and RBV were modelled to exhibit competitive binding to the same target site. Therefore, the overall inhibitory effect of FAV and RBV was described by a competitive binding model. If either the FAV (C_FAV_) or the RBV concentration (C_RBV_) was zero, this competitive binding model converges to a Hill equation for one drug. The equation for INH was split into three parts for easier representation. We fixed the maximum extent of inhibition (Imax_FAV_ and Imax_RBV_) both to 1.0 and estimated the FAV and RBV concentrations causing 50% of maximum inhibition (IC_50_FAV_ or IC_50_RBV_):(15)$${\mathrm{Eff}}_{\mathrm{FAV}}=\left(\frac{C_{\mathrm{FAV}}}{{\mathrm{IC}50}_{\mathrm{FAV}}}\right){\mathrm{Hill}}_{\mathrm{FAV}}$$
(16)$${\mathrm{Eff}}_{\mathrm{RBV}}=\left(\frac{C_{\mathrm{RBV}}}{{\mathrm{IC}50}_{\mathrm{RBV}}}\right){\mathrm{Hill}}_{\mathrm{RBV}}$$
(17)$$\mathrm{INH}=\frac{{\mathrm{Imax}}_{\mathrm{FAV}}\cdot {\mathrm{Eff}}_{\mathrm{FAV}}+{\mathrm{Imax}}_{\mathrm{RBV}}\cdot {\mathrm{Eff}}_{\mathrm{RBV}}}{{\mathrm{Eff}}_{\mathrm{FAV}}+{\mathrm{Eff}}_{\mathrm{RBV}}+1}$$

*Batch-to-batch variability of viral replication:* Our in-vitro experiments employed three cell lines and three drugs with multiple concentrations each. Batch to batch variability in viral replication kinetics was characterized using the no treatment controls. In all cell lines, we noticed significantly different viral replication profiles among batches of assays (Appendix A). To account for between batch variability, our MBM estimated different population means for the synthesis rate constant (k_syn_) and the infection rate constant (k_infect_) between batches.

*System outputs and residual error model:* In our in vitro experiments, viral burden was reported as plaque-forming units per mL (PFU/mL). Our model used log_10_ PFU/mL as the dependent variable. To account for samples below the limit of quantification, the Beal M3 method [25] was used as implemented in the S-ADAPT software. We used an additive residual error on log_10_ scale to fit the PFU/mL data.

*Parameter variability model:* The variability of parameters was described by an exponential variability model. When Imax parameters were estimated in earlier versions of this model, we constrained the individual Imax estimates between 0 and 1 via a logistic transformation. Normal distributions were also employed for parameters estimated on log-scale (i.e., for Log_U and Log_I). We implemented a variance burn then shrink routine as described previously [26] and fixed the between curve variability for all parameters to a final coefficient of variation 10%, since the experimental variability between replicates were small.

*Model qualification:* To assess the goodness of fit, the individual and population curve fit plots over time, and individual and population fits versus observation plots were employed. To evaluate the predictive performance, we utilized the normalized prediction distribution error (NPDE) plots. The best model was selected based on standard diagnostic plots, the objective function (negative log-likelihood in the S-ADAPT software), and the plausibility of pharmacodynamic (PD) parameter estimates.

*Simulations for the combined pharmacokinetic/pharmacodynamic (PK/PD) model:* The concentration-time profiles of FAV and IFN were simulated based on our previously published PK model [23]. For FAV, the clinically relevant dosage regimens against influenza [27] and Ebola [28] infections were used for PK simulations. Simulated PK profiles under mono- or combination therapies were used as input functions to predict the drug effects in HT-1080, HFF-1 and SK-N-MC cells separately, or in a combined system of all three cell lines. To account for the in-vivo situation, the number of host cells per cell line was maintained at 10^6.3^ cells/mL

*Software:* We used the Monte Carlo Parametric Expectation Maximization (MC-PEM [29]; also called importance sampling) algorithm in the parallelized S-ADAPT (version 1.57) software. The SADAPT-TRAN facilitator tool was employed [26,30]. Simulations were performed in the Berkeley Madonna (version 8.3.18) software [31]. The lattice package in the statistical programming language R (version 3.5.2) [32], and Graphpad Prism 7 were used for data visualization.

## 3. Results

### 3.1. Antiviral Evaluations of Broad-Spectrum Agents as Monotherapy

FAV was most effective in HT-1080 cells, yielding a clear exposure-response relationship that was maintained over the entire three-day experiment and resulting in an EC_50_ value of 41.92 µg/mL (Figure 1a). The antiviral effect of FAV was markedly less pronounced in SK-N-MC cells, in which concentrations ≤39.3 µg/mL yielded viral burden that was identical to that of the control (Figure 1b). FAV concentrations of 78.6 µg/mL and 157 µg/mL, suppressed the production of infectious CHIKV in this cell line by approximately 10-fold on day 2 post-infection; however, no effect was observed after one day of treatment. The EC_50_ value of FAV in SK-N-MC cells was >157 µg/mL. Finally, FAV was completely ineffective against CHIKV in HFF-1 cells, as viral burden in all treatment arms remained identical to the control group (Figure 1c). This lack of antiviral activity in HFF-1 cells is due to the fact that neither intracellular FAV nor FAV-RTP was detected in this cell line, indicating that the drug does not penetrate or is not retained within HFF-1 cells to exhibit activity (Appendix A). FAV and FAV-RTP levels were readily observed in both HT-1080 and SK-N-MC cells (Appendix A).

IFN had the greatest antiviral effect in HFF-1 cells, followed by HT-1080 cells, then SK-N-MC cells (Figure 1d–f). Infectious CHIKV was suppressed by IFN in HT-1080 cells in an exposure-dependent manner and markedly inhibited viral replication, with concentrations of 100 IU/mL and above resulting in at least a 1000-fold reduction (i.e., ≥3 log_10_ PFU/mL) in infectious viral burden. Moreover, the antiviral effect was maintained throughout the entire experiment (Figure 1d). The EC_50_ value for IFN against CHIKV was 21.7 IU/mL in HT-1080 cells. In SK-N-MC cells, concentrations of 100 IU/mL were required to inhibit CHIKV production; however, the antiviral activity of IFN at these concentrations was substantially less than the effect observed in HT-1080 and HFF-1 cells (Figure 1e). IFN appeared to slow down CHIKV replication, but not completely suppress it, as viral burden continued to increase throughout the duration of treatment in all experimental arms. IFN exhibited an EC_50_ value equivalent to 171 IU/mL in SK-N-MC cells. Finally, HFF-1 cells were most susceptible to the antiviral effects of IFN, yielding an EC_50_ value of 8.73 IU/mL. Suppression of viral replication was most pronounced at drug concentrations ≥10 IU/mL (Figure 1f). On day 2, IFN concentrations of 10 IU/mL reduced viral titers by 3 log_10_ PFU/mL, whereas 100 IU/mL and 1000 IU/mL provided similar levels of suppression resulting in a 5 log_10_ PFU/mL reduction in viral burden. An IFN concentration of 10,000 IU/mL yielded viral titers that were similar to the limit of detection for the plaque assay (Figure 1f).

The antiviral effect of RBV was most pronounced in HT-1080 cells, resulting in an EC_50_ value of 124 µg/mL. RBV concentrations ≥100 µg/mL were required to appreciably inhibit viral replication in this cell line (Figure 1g). Treatment with 100 µg/mL of RBV decreased viral burden by 1.33 log_10_ PFU/mL on day 2 when peak viral titers were achieved in the no treatment control; further inhibition of viral production was observed at the highest concentration of RBV evaluated (1000 µg/mL) which reduced viral titers by 3.38 log_10_ PFU/mL (Figure 1g). In SK-N-MC cells, RBV concentrations lower than 100 µg/mL were ineffective relative to the control and antiviral effect was observed in the 100 and 1000 µg/mL treatment arms. A concentration of 100 µg/mL RBV resulted in a transient reduction in viral burden that was lost by day 3 of treatment, and exposure at 1000 µg/mL RBV decreased viral titers by 3.98 log_10_ PFU/mL on day 2 (Figure 1h). RBV yielded an EC_50_ of 296 µg/mL against CHIKV in SK-N-MC cells. HFF-1 cells were least susceptible to RBV antiviral effect (EC_50_ > 1000 µg/mL). Drug concentrations ranging from 0.1 to 100 µg/mL did not successfully suppress viral replication as viral burden in these treatment arms was nearly identical to those of the no treatment control. The antiviral effect of RBV was only observed at 1000 µg/mL, this concentration yielded a 2.65 log_10_ PFU/mL decline in CHIKV replication on day 2 (Figure 1i). Cytotoxicity was observed at high RBV concentrations (≥100 µg/mL) in all three cell lines, and likely contributed to the decline of viral titers on day 3.

### 3.2. Antiviral Evaluations of Combination Therapy

For these studies, 6 concentrations of FAV (0 to 157 µg/mL) and IFN (0 to 10,000 IU/mL) were evaluated alone and in every possible combination of concentrations against CHIKV in HT-1080, SK-N-MC, and HFF-1 cells. HT-1080 cells were most susceptible to enhanced antiviral activity of FAV plus IFN combination regimens. CHIKV achieved a peak viral burden of 5.9 log_10_ PFU/mL on day 2 in this cell line (Figure 2a); combination therapy at clinically achievable concentrations of FAV (39.3 µg/mL) and IFN (100 IU/mL) yielded an approximate 50-fold reduction in CHIKV replication relative to the no treatment control. FAV at 157 µg/mL plus IFN at 10,000 IU/mL inhibited CHIKV by ~3.1 log_10_ PFU/mL while monotherapy at these concentrations yielded an approximate 2.5 log_10_ PFU/mL reduction in viral burden for both FAV and IFN (Figure 2a,d).

In SK-N-MC cells, IFN and FAV monotherapy inhibited viral replication in a concentration-dependent manner; however, CHIKV susceptibility to FAV was minimal in this cell line. Single agent therapy reduced viral burden by approximately 3.5 log_10_ PFU/mL at 10,000 IU/mL IFN and 1.3 log_10_ PFU/mL at 157 µg/mL FAV on day 2. Therapy with FAV plus IFN slightly augmented antiviral activity relative to monotherapy; IFN 10,000 IU/mL plus FAV 157 µg/mL inhibited CHIKV by approximately 4 log_10_ PFU/mL relative to the control arm. The clinically achievable regimen of 39.3 µg/mL FAV and 100 IU/mL IFN reduced viral titers by 1.6 log_10_ PFU/mL while FAV and IFN monotherapy at these concentrations inhibited CHIKV by 0.5 log_10_ PFU/mL and 0.9 log_10_ PFU/mL, respectively (Figure 2b,e).

FAV and IFN combination regimens did not enhance antiviral effect against CHIKV in HFF-1 cells as addition of FAV yielded CHIKV titers that were not different from those achieved by IFN monotherapy. IFN effectively inhibited CHIKV replication in a concentration dependent manner. Treatment with the clinically achievable concentration of 100 IU/mL IFN resulted in an approximate 4.4 log_10_ PFU/mL reduction in viral burden relative to the no treatment control, while maximal viral suppression of 5.2 log_10_ PFU/mL was observed at 10,000 IU/mL IFN (Figure 2c,f).

### 3.3. Mathematical Modeling

A pharmacokinetic/pharmacodynamic (PK/PD) model was developed to describe the relationship between production of infectious CHIKV over time and the inhibitory effect of FAV, RBV, and IFN on viral replication. This model also accounted for the cytotoxic effect associated with exposure to high RBV concentrations (Figure 3). The model was used to describe and predict viral burden for all mono- and combination therapy regimens within one cell line simultaneously. The curve fits were reasonably precise and unbiased (Figure 4) for both monotherapy (Figure 1) and combination regimens (Figure 2a–c).

The I (i.e., for the Ebola regimen) values for FAV, IFN, and RBV were fixed as 1 indicating essentially complete inhibition of viral replication at high drug concentrations. The IFN IC_50_ values were 0.0841 IU/mL in HFF-1 cells, 5.44 IU/mL in SK-N-MC cells, and 4.86 IU/mL in HT-1080 cells. The FAV IC_50_ estimates were 79.9 µg/mL in SK-N-MC cells and 24.6 µg/mL in HT-1080 cells (FAV did not penetrate HFF cells and, therefore, the FAV effect was not estimated in this cell line). The IC_50_ estimates for RBV were 664 µg/mL in HFF-1 cells, 196 µg/mL in SK-N-MC cells, and 294 µg/mL in HT-1080 cells (Table 1).

The mathematical model captured trends in antiviral activity of FAV and IFN combination regimens well. The combination of FAV plus IFN was predicted to yield substantial viral inhibition in HT-1080 and SK-N-MC cells (Figure 2). In HFF-1 cells, considerable activity was predicted for IFN monotherapy. Model predictions indicated therapy with the clinically achievable concentrations of 39.3 µg/mL FAV and 100 IU/mL IFN would markedly suppress CHIKV replication by approximately 2 log_10_ PFU/mL in HT-1080 cells and 1.5 log_10_ PFU/mL in SK-N-MC cells relative to the no treatment control arms (Figure 2).

### 3.4. Simulated Antiviral Activity of Favipiravir (FAV) and Interferon-Alpha (IFN) Combination Regimens against Chikungunya Virus (CHIKV)

Our results indicated combination regimens with FAV plus IFN were the most promising candidates for future study as they inhibited CHIKV replication in multiple cell lines at clinically achievable concentrations. The mathematical model was used to simulate antiviral activity of FAV and IFN mono- and combination therapy against CHIKV in the three cell lines evaluated. The PK profiles associated with clinically relevant FAV and IFN regimens were predicted to produce different extents of inhibition of viral replication that were considerably influenced by cell type. IFN simulations were based on the clinical regimen of 36 million international units (MIU) twice daily (BID) [33]. FAV simulations were based on two clinical regimens that have been previously described, the influenza regimen (1800 mg at 0 h and 12 h on day 1 followed by 800 mg BID) and the Ebola regimen (2400 mg at 0 h, 8 h, and 1600 mg at 16 h followed by 1200 mg BID starting on day 2) [16,23].

Overall, model simulations indicated that combination therapy with clinical regimens of FAV and IFN would considerably delay viral replication relative to the effect of monotherapy (Figure 5a). In HT-1080 cells, CHIKV viral burden reached a peak of 8.9 log_10_ PFU/mL in the absence of therapy. FAV monotherapy decreased viral burden relative to that of the control by 1.4 log_10_ PFU/mL for the influenza regimen, and by 4.0 log_10_ PFU/mL for the Ebola regimen. By day 10, IFN monotherapy almost completely inhibited viral replication decreasing viral burden by 8.3 log_10_ PFU/mL. Regimens containing FAV plus IFN completely suppressed CHIKV production and the degree of viral inhibition over the 10-day simulation was nearly identical for both combination regimens (Figure 5b). Excitingly, these model simulations predicted that the substantial antiviral effect achieved by mono- and combination regimens against CHIKV in this cell line would be sustained throughout the simulated 10-day course of therapy.

In SK-N-MC cells, peak viral titers of 8.5 log_10_ PFU/mL were achieved. CHIKV suppression from FAV monotherapy was slight for the influenza and Ebola regimens. IFN monotherapy markedly suppressed CHIKV replication at early time points but antiviral effect began to wane throughout the simulated course of therapy, and by day ten, predicted IFN effect was nearly attenuated. Simulations indicated combination regimens of FAV plus IFN yield greater viral suppression relative to that of monotherapy, especially at early time points. On day 2, the Ebola dose of FAV plus IFN decreased viral burden by 2.3 log_10_ PFU/mL relative to the control. Like IFN therapy, combinations delay but do not ultimately inhibit CHIKV replication as viral burdens begin to approach peak viral titers by day 5. At day 10, predicted viral suppression relative to the control was 0.3 log_10_ PFU/mL for the influenza FAV regimen and 0.4 log_10_ PFU/mL for Ebola FAV regimen (Figure 5c).

CHIKV achieved a peak viral burden of 8.4 PFU/mL in HFF-1 cells. Clinical regimens of FAV are predicted to be completely ineffective against CHIKV, as simulated viral burden for these regimens were identical to that of the control. IFN monotherapy extensively suppressed CHIKV, inhibiting CHIKV replication by up to 6.2 log_10_ PFU/mL relative to that of the control by Day 10. As there was no PD drug interaction between FAV and IFN in this cell line, combination regimens were predicted to be as effective as IFN monotherapy (Figure 5d).

## 4. Discussion

Due to the pantropic nature of CHIKV infection, it is imperative to identify antiviral treatment strategies that effectively inhibit viral replication in numerous types of tissue targeted by the virus. Here, we showed that FAV and IFN substantially inhibited CHIKV replication in multiple clinically relevant human cell lines at therapeutically feasible concentrations. We then evaluated the antiviral potential of these promising compounds in combination and developed a PK/PD model to describe and predict the effectiveness of clinical regimens of FAV and IFN when human drug-concentration time profiles associated with these agents are simulated.

In vitro evaluations and predictions made from our PK/PD model indicated that combination therapy with FAV and IFN enhanced viral suppression in HT-1080 and SK-N-MC cells relative to the effects of monotherapies. Moreover, the antiviral activity exhibited by these compounds was observed at clinically achievable drug concentrations. The addition of FAV to IFN did not result in an increase in anti-CHIKV activity on HFF-1 cells and antiviral effect was solely attributed to IFN. Clinically achievable concentrations of IFN resulted in substantial viral suppression in HFF-1 cells. It is important to note that the presence of FAV did not alter the ability of IFN to suppress CHIKV replication (Figure 2c), suggesting that FAV and IFN did not exhibit antagonism. Therefore, the addition of FAV did not pose a threat to negatively impact the effectiveness of IFN in HFF-1 cells, despite the fact that FAV does not exhibit antiviral activity against the virus in this cell type.

These findings are important, as this allows one to consider a new strategy for combination therapy against a virus that targets multiple tissue and cell types. Traditionally, combination antiviral therapy is utilized as a means to increase antiviral effect and/or prevent the emergence and spread of viruses that have a reduced susceptibility to either agent. The data presented here allow for one to think about using combination therapy beyond this traditional scope, and instead using a combination of drugs that will be delivered to infected cells so that at least one agent in that combination will demonstrate antiviral activity in that afflicted cell type (so long as antagonism is not observed). This is very important to consider for viral infections with a broad tissue and cellular tropism; thus, the degree of additivity/synergy or resistance suppression should not be the only virological outputs evaluated for combination therapy in these types of viral infection, although they are important and extremely desirable. Instead, broad-spectrum effectiveness in different cell types should also be carefully considered. Overall, these findings demonstrate the effectiveness of FAV and IFN against CHIKV in diverse cell types, strongly supporting further investigation of this combination regimen as a treatment strategy for CHIKV.

Our in vitro antiviral experiments were performed using static drug conditions for each agent, meaning the concentration of drug was constant for the duration of the experiment. This experimental setting does not mimic the clinical scenario, as plasma concentration-time profiles for antiviral agents are dynamic following administration and fluctuate according to each drug’s specific PK profile. We leveraged our refined PK/PD mathematical model to conduct simulations using the PK profiles associated with clinical regimens of FAV and IFN in combination to predict the overall anti-CHIKV activity of these regimens in humans. The simulations showed that, generally, the combination regimens inhibited the production of infectious CHIKV over a 10-day period. The lone exception occurred in HFF-1 cells, for which IFN containing regimens yielded identical degrees of viral suppression since IFN was the only agent driving antiviral effect in this cell line. Overall, IFN in combination with the Ebola FAV regimen resulted in the maximum extent of viral inhibition in all cell lines. This combination often delayed viral replication, slowing the replication process, but did not completely suppress it based on our in vitro data. However, suppression was achieved in the HT-1080 cells with the predicted viral burden reaching undetectable levels after 8 days of therapy in both combination regimens evaluated. Being that the HT-1080 cells represent connective tissue, these findings are encouraging since the major disease manifestation of CHIKV infection is persistent and debilitating joint pain.

Notably, our studies and simulations do not take into account the human immune system. It is likely that our analyses may underestimate the clinical antiviral activity of the regimens under evaluation, as additional suppression is expected to be achieved in the face of a functional immune response (which is lacking in vitro). The implications of this are two-fold. First, the delay in viral replication by the effective treatment regimens may give enough time to allow for the immune system to clear the virus when titers are blunted. Therefore, suppression (complete or not) by antiviral therapy may not be required for a positive outcome. Second, lower doses of either IFN, FAV or both may be employed without compromising effectiveness when a functional immune system is considered.

There are some drawbacks associated with FAV and IFN as combination therapy, in spite of the promising antiviral activity observed with this regimen. These limitations have been discussed previously [23]. Briefly, FAV cannot be administered to pregnant women due to the fact that it has shown teratogenic and embryotoxic effects in various animal model systems [34]. Therefore, this treatment strategy would not be available to those who are pregnant or are trying to become pregnant. Another limitation is that the degree of penetration into the central nervous system is not known for FAV and is restricted for IFN [35]. Such penetration issues may reduce the utility of the combination regimen as a treatment strategy for the neurological consequences of CHIKV infection. However, this limitation may be overcome via alternative administration routes. Controlling the viral infection in the peripheral tissues, like the skin and connective tissue, may prevent or decrease neurological disease manifestations and render this hurdle a non-issue.

There are several limitations to this study. First, antiviral activity of FAV and IFN was evaluated against one strain of CHIKV. To conduct these studies in a biosafety level 2 (BSL-2) setting, the vaccine strain of CHIKV was selected for preclinical drug evaluations. Future experiments will evaluate the antiviral effect of these agents against other clinically relevant CHIKV strains. Second, our in vitro evaluations were conducted using static drug concentrations which do not accurately reflect the concentration time profiles observed in humans following administration of an antiviral agent. Simulation of human PK profiles associated with clinical FAV and IFN combination regimens using the hollow fiber infection model system will allow us to more closely predict the antiviral effect of these regimens under dynamic drug concentrations. These data can then be used to experimentally validate the mathematical model predictions.

In conclusion, our results suggested that combination therapy of FAV and IFN holds promise as a treatment strategy against CHIKV. Moreover, these findings suggest an alternative approach for designing and rationally optimizing combination regimens for pantropic viruses, such as CHIKV. This alternative approach does not focus solely on achieving enhanced antiviral activity or resistance suppression, but also considers combining agents that have documented effectiveness in distinct tissues or cell types relevant to viral infection. This will allow for at least one of the agents in the combination to effectively inhibit viral replication in the virally-targeted tissue or cell, maximizing antiviral activity in ideally all of the afflicted tissues. To our knowledge, this has not been previously explored for antivirals in combination.

## Figures and Tables

**Figure 1 microorganisms-09-00307-f001:**
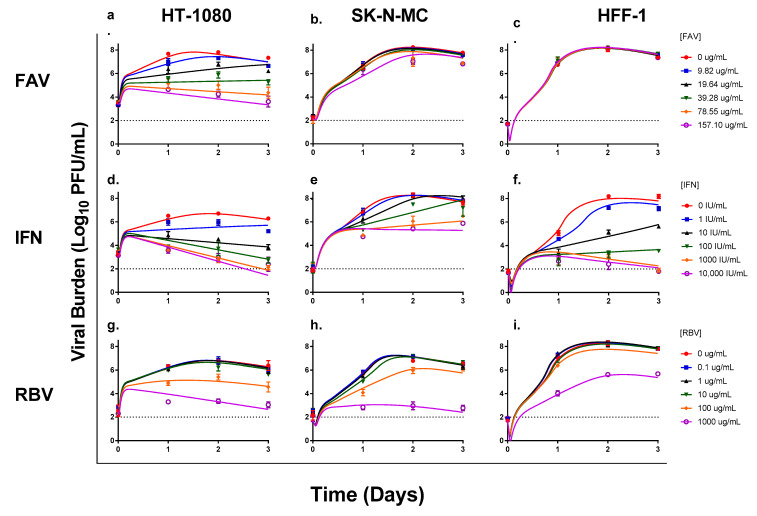
Antiviral activity of favipiravir (FAV), interferon-alpha (IFN), and ribavirin (RBV) monotherapy against Chikungunya virus (CHIKV) in human connective tissue (HT-1080), neurons (SK-N-MC), and skin (HFF-1) cells. HT-1080 (**a**,**d**,**g**), SK-N-MC (**b**,**e**,**h**), and HFF-1 (**c**,**f**,**i**) cells were infected at multiplicities of infection of 0.1, 0.01, and 0.001, respectively. Infected cells were treated with different concentrations of FAV, IFN, or RBV. Viral burden was quantified via plaque assay on Vero cells, and reported as log_10_ plaque-forming units/mL (PFU/mL). Data points represent the mean of three independent observations and error bars correspond to one standard deviation (for some observations, the error bars were smaller than the markers). The lines through the data points signify the individually fitted viral burden as determined by the mathematical model. The dashed line signifies the assay limit of detection.

**Figure 2 microorganisms-09-00307-f002:**
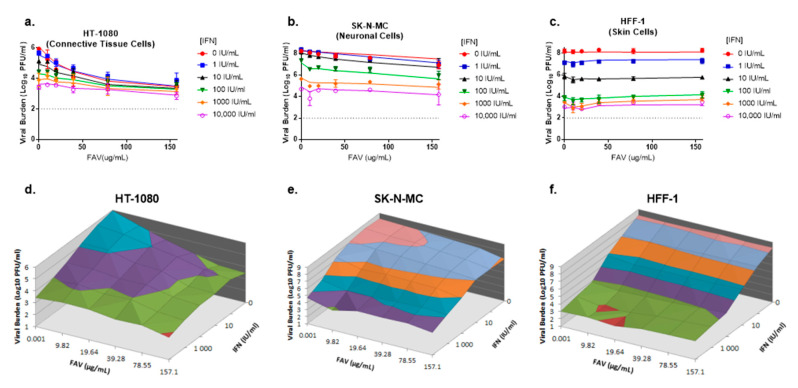
Antiviral activity of favipiravir (FAV) and interferon (IFN) combination therapy against CHIKV in HT-1080, SK-N-MC, and HFF-1 cells (**a**–**c**). HT-1080 (**a**), SK-N-MC (**b**), and HFF-1 (**c**) cells were infected with CHIKV at MOIs of 0.1, 0.01, and 0.001, respectively. Different concentrations of FAV, IFN or both were added to cells following infection. Day 2 viral burden was quantified via plaque assay on Vero cells. Data points represent the mean of three independent samples and error bars correspond to one standard deviation. Lines through the data points signify the predicted viral burden as determined by the mathematical model. The dashed line signifies the assay limit of detection. (**d**–**f**) Three-dimensional plots illustrating viral burden for FAV + IFN combination regimens on HT-1080 (**d**), SK-N-MC (**e**), and HFF-1 (**f**) cells are shown.

**Figure 3 microorganisms-09-00307-f003:**
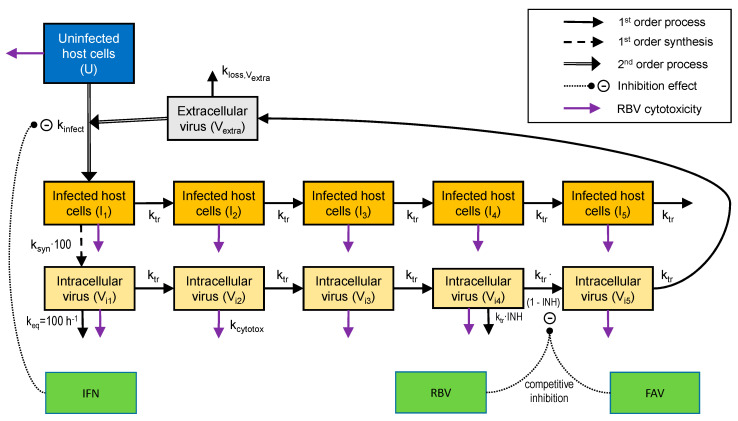
Structural model describing extracellular CHIKV infecting host cells, the production of intracellular virus by infected cells, as well as the inhibitory effects of FAV, IFN and RBV. The binding of extracellular virus to uninfected host cells was not affected by IFN. However, IFN inhibited the subsequent transformation from uninfected to infected host cells. Both infected host cells and intracellular virus were described by a series of five transit compartments each. In this model, virus was synthesized by the first infected host cell compartment. To achieve a rapid equilibrium between host cells in stage 1 and intracellular virus in stage 1, the first-order virus synthesis rate constant was multiplied by 100 and an equilibration rate constant (k_eq_ = 100 h^−1^) was incorporated for V_i1_. Both FAV and RBV inhibited the transition of intracellular virus between the fourth and the fifth intracellular virus compartment via a competitive mechanism. In addition, the cytotoxicity effect of RBV was incorporated in to the model and affected both uninfected and infected host cells, and indirectly also immature intracellular virus as described in the methods. In HFF-1 cells, FAV displayed poor intracellular permeability and did not contribute any inhibitory effect. However, FAV permeability was considerably better in the other cell lines.

**Figure 4 microorganisms-09-00307-f004:**
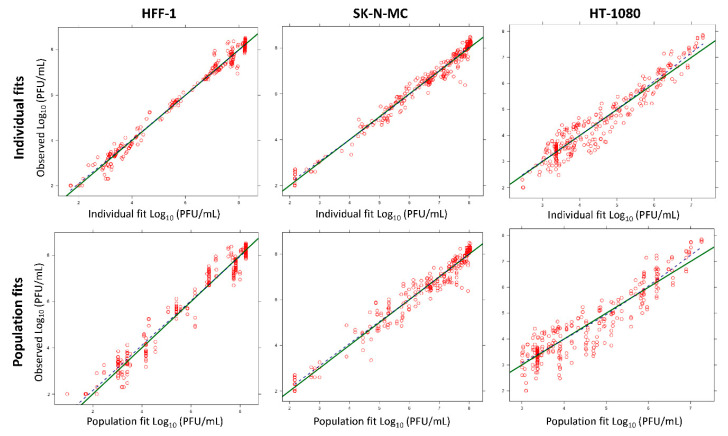
Predicted vs. observed fits for CHIKV viral burden in HT-1080, SK-N-MC, and HFF-1 cells. The green line represents the line of identity and the dashed blue line is a LOESS (locally weighted smoothing) smoother of the observations. Curve fits were unbiased, since the blue line fell on top of the green line.

**Figure 5 microorganisms-09-00307-f005:**
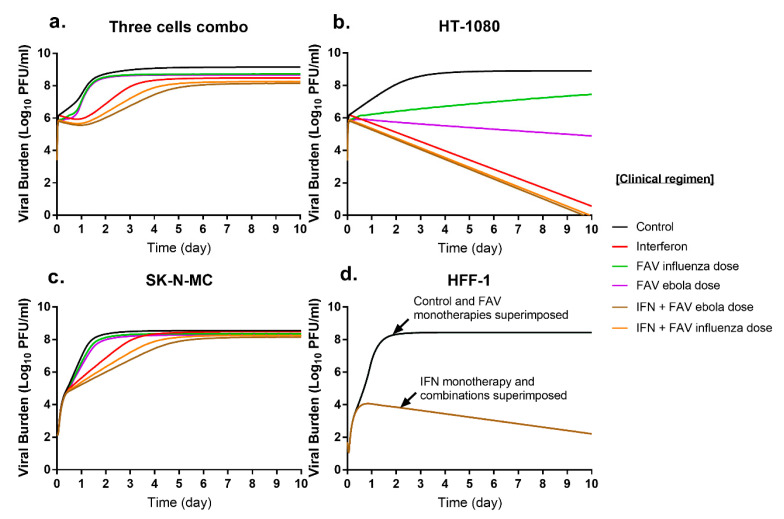
Simulated antiviral activity of clinically relevant FAV and IFN regimens (**a**) in HT-1080 (**b**), SK-N-MC (**c**), HFF-1 (**d**) cells. The PK profiles associated with the IFN clinical regimen of 36 million international units (MIU) twice daily (BID) were used for simulations. FAV influenza regimen of 1800 mg at 0 h and 12 h on day 1 followed by 800 mg BID; and FAV Ebola regimen of 2400 mg at 0 h, 8 h, and 1600 mg at 16 h followed by 1200 mg BID starting on day 2 were utilized for FAV simulations.

**Table 1 microorganisms-09-00307-t001:** Population mean parameter estimates for the pharmacokinetic/pharmacodynamic (PK/PD) model of FAV, RBV and IFN antiviral activity against CHIKV.

Parameter Name	Symbol	Unit	HFF-1	SK-N-MC	HT-1080
Log_10_ of 2nd order infection rate constant for batch with fast growth	Log_10_ (k_infect_) fast	-	−6.17 (2.4%)	−7.41 (2.0%)	−8.53 (2.6%)
Log_10_ of 2nd order infection rate constant for batch with slow growth	Log_10_ (k_infect_) slow	-	−5.56 (1.5%)	−6.09 (13.7%)	−7.91 (1.3%)
Synthesis rate constant for intracellular virus (batch with fast growth)	k_syn_ fast	1/h	221 (9.0%)	100 (10.7%)	31.3 (20.3%)
Synthesis rate constant for intracellular virus (batch with slow growth)	k_syn_ slow	1/h	137 (11.6%)	14.5 (13.5%)	5.83 (14.3%)
Mean delay time until release of intracellular virus in absence of drug; equivalent to the mean survival time of infected host cells	T_Delay_ = 5/k_tr_	h	22.4 (4.0%)	11.7 (9.5%)	0.965 (55.7%)
Mean survival time for extracellular virus	MST_Virus_ = 1/k_loss,virus_	h	14.4 (8.3%)	21.1 (10.8%)	18.7 (5.9%)
IFN concentration causing 50% of Imax	IC_50,IFN_	IU/mL	0.0841 (21.5%)	5.44 (36.9%)	4.86 (20.3%)
FAV concentration causing 50% of Imax	IC_50,FAV_	ug/mL	n/a	79.9 (11.3%)	24.6 (16.3%)
RBV concentration causing 50% of Imax	IC_50,RBV_	ug/mL	664 (5.8%)	196 (8.7%)	294 (14.4%)
Hill function for IFN	Hill_IFN_	-	0.725 (4.1%)	0.558 (4.4%)	0.902 (61.2%)
Hill function for FAV	Hill_FAV_	-	n/a	0.784 (18.9%)	1.04 (12.8%)
Hill function for RBV	Hill_RBV_	-	1.53 (12.8%)	0.917 (14.5%)	1.08 (30.6%)
Maximum cytotoxicity rate constant by RBV	kmax_RBV_	1/h	0.1 (fixed)	0.1 (fixed)	0.1 (fixed)
RBV concentration causing 50% of RBV related cytotoxicity	SC_50,RBV_	ug/mL	150 (fixed)	150 (fixed)	150 (fixed)
Log_10_ of initial concentration of uninfected cells (cells/mL)	Log_U	-	6.30 (fixed)	6.30 (fixed)	6.30 (fixed)
Log_10_ of initial concentration of infected cells (IC_I1_ in cells/mL)	Log_I	-	3.22 (2.3%)	3.37 (4.8%)	4.73 (1.8%)
Initial condition for extracellular virus	IC_Vextra_	PFU/mL	50 (fixed)	148 (9%)	2,330 (9.5%)
SD of additive error for viral load on log_10_ scale	SDin	-	0.307 (5.6%)	0.318 (6.7%)	0.420 (5.4%)

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
