# Peer review of "Combination Regimens of Favipiravir Plus Interferon Alpha Inhibit Chikungunya Virus Replication in Clinically Relevant Human Cell Lines"

_microorganisms, 2021, doi:10.3390/microorganisms9020307_

Round 1

Reviewer 1 Report

All of my concerns for the previous version of the manuscript have been satisfied hence I suggest accepting this manuscript for publication.

Reviewer 2 Report

The authors have addressed my previous comments to my satisfaction. I recommend the manuscript be published.

Reviewer 3 Report

The authors have addressed most of the questions raised by the reviewers and the quality of the current manuscript has been improved. The reviewer would recommend the acceptance of this manuscript, although the novelty of this article is incremental.

This manuscript is a resubmission of an earlier submission. The following is a list of the peer review reports and author responses from that submission.

Round 1

Reviewer 1 Report

This manuscript describes experiments examining favipiravir and interferon combination therapy against Chikungunya virus in three different cell lines. A mathematical model is also used to create more pharmacodynamically realistic simulations of combination therapy. Overall, the manuscript has merit, but some details of the experiments and modeling are lacking.

  1. Why was Day 2 chosen for generating the experimental dose-response curves? Modeling shows that for at least two of the cell lines, the anitviral effects start to disappear if you go out a few more days. Are the experiments perhaps giving a misleading evaluation of the treatment efficacy by using such an early time point?
  2. While the manuscript repeatedly claims that FAV has no effect on HFF-1 cells, careful examination of the dose response curves (Fig. 1) shows a increased response at low doses of FAV particularly when IFN doses are high. Do the authors know what is going on there?
  3. How did the authors decide on the model formulation of the mechanism of action of the antivirals? The effect of IFN in particular has been modeled in a number of different ways (Pawelek et al. 2012, Plos Comp. Biol., for example). Similarly, what is the justification of the choice of favipiravir acting on the transition from compartment 4 to 5 --- why not any of the other compartments? Antiviral mechanism of action can change the predicted outcome of the combination therapy (Melville et al. 2018, Frontiers of Pharmacol.).
  4. The little check and cross at the bottom of Fig. 2 are confusing. Does this mean that the effect of FAV was modeled differently in the HFF-1 cell lines?
  5. Have the authors looked at the identifiability of all the estimated parameters? Maximum drug efficacy in particular is often not identifiable and will tend towards its upper boundary.

Author Response

Reviewer #1:

This manuscript describes experiments examining favipiravir and interferon combination therapy against Chikungunya virus in three different cell lines. A mathematical model is also used to create more pharmacodynamically realistic simulations of combination therapy. Overall, the manuscript has merit, but some details of the experiments and modeling are lacking.

  1. Why was Day 2 chosen for generating the experimental dose-response curves? Modeling shows that for at least two of the cell lines, the anitviral effects start to disappear if you go out a few more days. Are the experiments perhaps giving a misleading evaluation of the treatment efficacy by using such an early time point?

Response:  Day 2 was chosen as this was the time point in which viral burden peaked in the no-treatment control arm.  After day 2, viral burden decreases in the control arm due to the killing off of target cells from viral infection.  When there are no longer any target cells to propagate the infection, the virus in the media loses infectivity naturally due to the incubation conditions at 37°C.  Thus, choosing a later day may skew our results and alter our interpretations due to the decrease in viral burden in the control.  Additionally, for all the experimental conditions described in this paper, viral titers in the arms exposed to drug therapy are similar between days 2 and 3.  Therefore, we believe it is unlikely that choosing day 2 is misrepresenting our results. We have optimized this time point in previous projects (years go) and found day 2 to provide the most useful information.

  1. While the manuscript repeatedly claims that FAV has no effect on HFF-1 cells, careful examination of the dose response curves (Fig. 1) shows a increased response at low doses of FAV particularly when IFN doses are high. Do the authors know what is going on there?

Response:  Figure 2C shows combination therapy with FAV and IFN on HFF-1 cells.  The colored lines, representing different concentrations of IFN, are nearly straight across, indicating that the addition of FAV to IFN does not enhance antiviral activity and that any activity is solely driven by the IFN.  There is some “bounce” in the lines, but this is due to the nature of the plaque assay.  The plaque assay is a bioassay which will naturally give some variability since it is quantifying live virus.  The error bars on these data points signify this variability.  These viral titers are not different from each other, demonstrating that the addition of FAV does not enhance anti-CHIV activity of IFN.

  1. How did the authors decide on the model formulation of the mechanism of action of the antivirals? The effect of IFN in particular has been modeled in a number of different ways (Pawelek et al. 2012, Plos Comp. Biol., for example). Similarly, what is the justification of the choice of favipiravir acting on the transition from compartment 4 to 5 --- why not any of the other compartments? Antiviral mechanism of action can change the predicted outcome of the combination therapy (Melville et al. 2018, Frontiers of Pharmacol.).

Response:  Traditionally, IFN has been thought to induce an antiviral state in cells, making these cells refractory to infection.  Therefore, we chose to model the mechanism of action of IFN as blocking cells from viral infection.  We considered IFN exerting its pharmacology effect via inhibiting the virus getting in touch with and thereby infecting the host cells. Pawelek et al. 2012, Plos Comp. Biol presented a similar approach that IFN introduced refractory cells from total cells thereby limiting the cells that are available for infection. These two models used a slightly different mathematical representation, but both models implement the inhibitory effect of IFN extracellularly (i.e. IFN does not need to penetrate the cell membrane). With respect to FAV, the drug exerted the pharmacology effect in intracellular space, and inhibit the viral replication and maturation in infected host cells. From this perspective, the FAV effect can be placed in any intracellular compartment. We select the compartment 4 to 5 to keep consistence with our previous published Zika virus model. We found that the proposed mechanisms of drug action gave us good model performance for the drug combinations and therefore chose this model.

  1. The little check and cross at the bottom of Fig. 2 are confusing. Does this mean that the effect of FAV was modeled differently in the HFF-1 cell lines?

Response:  FAV displayed no (very limited) effect in HFF-1 cells because FAV showed poor permeability in this cell line. We used a red cross in Fig.2 to demonstrate the lack of permeability for FAV in HFF-1 cells. We have removed these items and placed a description instead in the figure legend. In the model, the IC50 of FAV could not be estimated for the HFF-1 dataset. Therefore, FAV did not exert a pharmacology effect in HFF-1 cells. When reporting the parameters, we reported NA for Imax, IC50 and Hill for FAV in HFF-1 cells.

  1. Have the authors looked at the identifiability of all the estimated parameters? Maximum drug efficacy in particular is often not identifiable and will tend towards its upper boundary.

Response:  We reported the parameter uncertainty in the modeling Table. The relative standard errors (SE%) or confidence intervals (for Imax) were tight indicating adequate precision of parameter estimates. Imax was estimated to be close to 1.0 displaying near-complete inhibition of the respective process in the model. This facilitated estimation of the IC50. We tested both models with Imax fixed to 1.0 and Imax estimated. The models behaved similarly. To provide consistency with previous models on other viral pathogens, we preferred to estimate Imax, since Imax is sometimes lower (e.g. 0.99 or 0.97). Estimating Imax in the present dataset shows that this process can be near-completely inhibited. RBV exerted both a pharmacology and a cytotoxicity effect. We fixed the RBV cytotoxicity parameters based on literature (a previous Zika virus paper from our group). This allowed us to readily estimate the parameters associated with RBV pharmacology effect.

Reviewer 2 Report

Review of the manuscript “Combination Regimens of Favipiravir plus Interferon 2 Alpha Inhibit Chikungunya Virus Replication in 3 Clinically Relevant Human Cell Lines” for MDPI journal Microorganisms.

In this manuscript, authors have explored the possibility of using Favipiravir (FAV) and Interferon alpha (IFN) in monotherapy or in combination therapy, using antiviral assays in conjunction with mathematical modelling and simulations. Authors have used the three different cell lines: HT-1080 (connective tissue cells), neuronal cell line SK-N-MC and skin cell derived HFF-1 cells. Authors show that combination of FAV and IFN enhanced antiviral activity in HT-1080 and SK-N-MC cells, but 22 not in HFF-1 cells. Authors have developed a “pharmacokinetic/pharmacodynamics model” and ran simulations using it that predict relevant concentrations of FAV plus IFN completely which suppressed CHIKV replication in HT-1080 25 cells, and reduced titers in SK-N-MC cells.

This work is very similar to the work done by authors on Zika virus using the same drugs: Favipiravir, Interferon alpha and Ribavirin. Thus, this work lacks novelty but provides important data for developing antivirals against CHIKV.

Authors should explain their rationale for using the “mathematical model” for predicting optimal antiviral concentrations in their assays accompanied by a short summary of the model.

Authors have used very confusing language at some places to explain the terms in building the model. E.g. “Host cell dynamics: Extracellular chikungunya virus (Vextra) infected uninfected host cells (U) to form the initial infected host cells (Ii1).” P3L102.

Figure legend accompanying the figure2 is inadequate in explaining the complicated model.

Labels on Figure 4 are not readable.

Figure 5 is labelled Figure 4.

Figure4 and 5 can be combined as panels of the same figure.

Author Response

Reviewer #2: In this manuscript, authors have explored the possibility of using Favipiravir (FAV) and Interferon alpha (IFN) in monotherapy or in combination therapy, using antiviral assays in conjunction with mathematical modelling and simulations. Authors have used the three different cell lines: HT-1080 (connective tissue cells), neuronal cell line SK-N-MC and skin cell derived HFF-1 cells. Authors show that combination of FAV and IFN enhanced antiviral activity in HT-1080 and SK-N-MC cells, but 22 not in HFF-1 cells. Authors have developed a “pharmacokinetic/pharmacodynamics model” and ran simulations using it that predict relevant concentrations of FAV plus IFN completely which suppressed CHIKV replication in HT-1080 25 cells, and reduced titers in SK-N-MC cells.

  1. This work is very similar to the work done by authors on Zika virus using the same drugs: Favipiravir, Interferon alpha and Ribavirin. Thus, this work lacks novelty but provides important data for developing antivirals against CHIKV.

Response:  We are very delighted to see that the reviewer acknowledged the importance of this work.  Although the work done with Zika virus is similar, it is not identical, as the antiviral effect of these drugs yield different outcomes for CHIKV than for Zika.  Moreover, it is the first time these agents have been evaluated simultaneously on these cell lines, cell lines that are clinically relevant to CHIKV infection.  Finally, the model, although similar, was amended to study CHIKV.  Thus, we agree with the reviewer that these data are important for the development of antivirals against CHIKV.

  1. Authors should explain their rationale for using the “mathematical model” for predicting optimal antiviral concentrations in their assays accompanied by a short summary of the model.

Response:  We added the following paragraph to the discussion:

The proposed mathematical model accounted for the mechanisms of action for the drugs testes. This approach allowed us to adequately characterize the drug efficacy. Of note, RBV displayed considerable cytotoxicity at concentration greater than 100 µg/mL. The cytotoxicity would lead to a significant overestimation of the inhibitory effect by RBV. The proposed model could distinguish between the pharmacological and cytotoxic effects of RBV. Moreover, we implemented the clinical pharmacokinetics of IFN, FAV and RBV, and integrated the dynamic PK simulation into our simulations. This allowed us to predict the outcomes under clinically relevant drug concentration time profiles in humans. Finally, CHIKV has demonstrated a broad viral tropism in vivo as multiple tissue types are targeted during human infection. Our model integrated different cell lines into one system and thereby provides a more comprehensive evaluation of antiviral efficacy.

  1. Authors have used very confusing language at some places to explain the terms in building the model. E.g. “Host cell dynamics: Extracellular chikungunya virus (Vextra) infected uninfected host cells (U) to form the initial infected host cells (Ii1).” P3L102.

Response:  We deleted the subheading “host cell dynamics”, since it caused confusion. The extracellular virus was denoted as Vextra in the model and equations. Therefore, we cannot change this name and we believe this is an adequate symbol for this term in the model as shown in the model structure diagram.

  1. Figure legend accompanying the figure2 is inadequate in explaining the complicated model.

Response:  We revised the Figure caption as follows:

Figure 3: The structural model describing extracellular CHIKV infecting the host cells, the production of infectious virus from infected cells, as well as the inhibitory effect of FAV, IFN and RBV on viral replication. IFN inhibited the conversion of uninfected host cell by extracellular virus to form infected host cell. Both FAV and RBV inhibited the transition of intracellular virus between the 4th and the 5th intracellular virus compartment. In addition, the cytotoxicity effect of RBV was accounted in the model. In HFF-1 cells, FAV displayed poor intracellular permeability; however, its permeability in the other cell lines was considerably better.

  1. Labels on Figure 4 are not readable.

Response:  This has been fixed.  Thank you for the edit suggestion.

  1. Figure 5 is labelled Figure 4.

Response:  This has been fixed.  Thank you for the edit suggestion.

  1. Figure4 and 5 can be combined as panels of the same figure.

Response:  We respectfully disagree with this reviewer.  We believe that figure 4 and figure 5 are better suited as separate figures.  Figure 4 shows the unbiased and precise nature of the model at predicting the viral burden during therapy, whereas Figure 5 shows the simulation results which predict the effectiveness of the dosage regimens against CHIKV when human pharmacokinetic profiles associated with the clinical regimens of FAV and IFN are simulated.

Round 2

Reviewer 1 Report

While the authors responded to my questions in their author response, the only change in the manuscript in response to my comments is altering one figure. Much of the comments they included in the author response need to be incorporated into the manuscript.

The authors adequately responded to all my comments (provided responses are incorporated into the manuscript) except for my question about parameter identifiability. Tight confidence intervals do not mean that a parameter is identifiable. Additionally, not all of their confidence intervals are all that great (I see some parameters with >50% error). The authors need to look at likelihood profiles and correlation plots to assess whether the parameters are identifiable.